# Case finding of tuberculosis among mining communities in Ghana

**Sally-Ann Ohene**[1]*, **Frank Bonsu**[2], **Yaw Adusi-Poku**[2], **Francisca Dzata**[2], **Mirjam Bakker**[3]

**1** World Health Organization Country Office, Accra, Ghana, **2** National Tuberculosis Control Program, Ghana Health Service, Accra, Ghana, **3** KIT Royal Tropical Institute, Global Health Amsterdam, Amsterdam, The Netherlands

* salohene@yahoo.com

## Abstract

### Background

Data on active TB case finding activities among artisanal gold mining communities (AMC) is limited. The study assessed the yield of TB cases from the TB screening activities among AMC in Ghana, the factors associated with TB in these communities and the correlation between the screening methods and a diagnosis of TB.

### Methods

We conducted secondary data analyses of NTP program data collected from TB case finding activities using symptom screening and mobile X-ray implemented in hard to reach AMC. Yield of TB cases, number needed to screen (NNS) and the number needed to test (NNT) to detect a TB case were assessed and logistic regression were conducted to assess factors associated with TB. The performance of screening methods chest X-ray and symptoms in the detection of TB cases was also evaluated.

### Results

In total 10,441 people from 78 communities in 24 districts were screened, 55% were female and 60% (6,296) were in the aged 25 to 54 years. Ninety-five TB cases were identified, 910 TB cases per 100,000 population screened; 5.6% of the TB cases were rifampicin resistant. Being male (aOR 5.96, 95% CI 3.25–10.92, P < 0.001), a miner (aOR 2.70, 95% CI 1.47–4.96, P = 0.001) and age group 35 to 54 years (aOR 2.27, 95% CI 1.35–3.84, P = 0.002) were risk factors for TB. NNS and NNT were 110 and 24 respectively.; Cough of any duration had the strongest association with X-ray suggestive of TB with a correlation coefficient of 0.48. Cough was most sensitive for a diagnosis of TB; sensitivity of 86.3% (95% CI 79.4–93.2) followed by X-ray, sensitivity 81.1% (95% CI 71.7–88.4). The specificities of the symptoms and X-rays ranged from 80.2% (cough) to 97.3% (sputum).

### Conclusion

The high risk of TB in the artisanal mining communities and in miners in this study reinforces the need to target these populations with outreach programs particularly in hard to reach

**Data Availability Statement:** All relevant data are within the paper and its Supporting Information files.

**Funding:** The authors received no specific funding for this work.

**Competing interests:** The authors have declared that no competing interests exist.

areas. The diagnostic value of cough highlights the usefulness of symptom screening in this population that may be harnessed even in the absence of X-ray to identify those suspected to have TB for further evaluation.

## Introduction

Tuberculosis (TB) remains the leading cause of death from an infectious agent in the twenty first century despite being curable with antimicrobial drugs [1]. An estimated 10 million people developed TB in 2019, while about 1.4 million deaths occurred as a result of the disease [1]. The WHO African Region has the highest TB incidence rate of 226 per 100,000 population [1]. Ghana classified a high HIV/TB burden, like several countries in the African region, grapples with TB control activities [2]. The TB cases notified annually in Ghana averaged 14, 500 TB cases in the years preceding the national prevalence survey conducted in 2013 and thereafter [2,3]. The prevalence survey findings however suggested that people being detected with TB represent just about 21% of actual cases highlighting the need to adopt different strategies to identify more cases [2]. The END TB Strategy of the World Health Organization emphasizes early diagnosis of TB that involves systematic screening of contacts and high-risk groups, universal drug–susceptibility testing and standardized treatment of TB and drug-resistant TB forms [4]. Early case detection coupled with a high treatment success leads to a cure of infectious TB cases and cuts the risk of transmission ultimately reducing the burden of TB. This has led to several countries adopting various strategies to improve case detection [5–7].

Ghana is one such country in which a 2013 comprehensive review of the Ghana National TB Program highlighted the need to improve access to TB prevention, treatment, care and support services for key affected populations including PLHIV, children, diabetics, and miners or those exposed to silica [8]. Consequently, the National TB Health Sector Strategic Plan for Ghana (2015–2020) clearly identified TB case detection among mining populations as one of these key populations for intervention [2].

High risk populations such as miners especially those engaged in illegal mining activities may however be in hard to reach areas where they may encounter challenges accessing health services including those for diagnosing TB [9,10]. Strategies such as deployment of mobile teams proved to be useful in identifying people with TB in such hard to reach populations and mining communities [10–12]. The implementation of active case finding (ACF) activities using mobile teams in hard to reach mining communities in Myanmar identified significantly higher TB prevalence in these townships than in the rest of the country [11]. In a study conducted among South African miners, death from TB was more than 2 times higher among those with TB diagnosed through passive case finding than those diagnosed through active screening exercises [13].

The Ghana NTP with funding from the Global Fund supported interventions to improve TB case finding among artisanal gold mining communities in the country [14]. Given the limited data on active TB case finding activities among small scale mining communities in Ghana, the aim of the study was to assess the yield of TB cases from the TB screening activities among artisanal mining communities, the factors associated with TB in these communities and the correlation between the screening methods and a diagnosis of TB.

## Methods

### Study design and setting

The study was cross-sectional consisting of secondary data analyses of routine NTP program data collected from TB case finding activities implemented in selected artisanal mining

communities in which mining activities are generally operated without license from authorities. These activities were undertaken from May 2017 to January 2018 as part of NTP's efforts to make available TB screening, testing and treatment to vulnerable populations in these hard to reach artisanal mining areas [2]. The communities were deemed to be at high risk for TB but with limited access to TB services. Having already undertaken passive TB control activities in artisanal mining areas in Ashanti, Brong Ahafo and Western regions (collective estimated population of 11,200,955 in 2017), the NTP in consultation with the respective Regional Health Directorates (RHD) selected 3, 6 and 12 districts respectively from each region with intense artisanal mining activities.

These 21 districts consisted of an estimated total population of 2,120,660 in 2017 with TB notification rates ranging from 37 to 394/100,000 population. In consultation with the RHD and District Health Directorates (DHD), 78 communities having artisanal mining activities, were selected from these districts for TB screening. In these communities, unlicensed gold extraction using crude mining extraction methods along terraces, mineral deposits in water ways and underground mines are undertaken by small-scale artisanal miners (locally known to as *galamsey)* [15,16]. These unregulated mining activities usually occur in hard to reach areas with limited access to health services. The communities were thus selected on the basis of their vulnerability to TB and being in hard to reach areas.

The conduct of the TB screening exercise was similar in the respective communities in which they were conducted across the three regions. The NTP deployed mobile teams consisting of a team leader to superintend the screening team, field coordinator to coordinate the field activities, radiographer to operate the digital X-ray machine, a physician-assistant to read the digital X-ray information technologist to maintain network connectivity for the transmission and storage of X-ray images and data, engineer to maintain the X–ray equipment and electrical generators, support team members to conduct screening and collect sputum samples and drivers for the digital X-ray van and vehicle to transport the team and other logistics. A week before the arrival of the team to the community, the district TB focal person in collaboration with the community volunteers undertook community mobilization using various methods including house to house sensitization and community radio announcements informing the community member of the upcoming free TB screening exercise, the date and the place where the team will be located.

**Study population and data collection.** The TB screening exercise was offered purely on voluntary basis to all community members 15 years and above who were willing to participate. Sampling or randomization of the population was therefore not done and neither was enumeration of community members done before the exercise. All community members 15 years and above who voluntarily made themselves available in the designated communities were eligible to be included in the screening exercise though the dataset showed that a handful of children less than 15 years were also screened. On the day of the exercise, community members voluntarily presenting themselves at the location of the mobile team were registered and their socio-demographic information recorded. A TB symptom screening was administered using a questionnaire inquiring about the presence or absence of cough, fever, sputum production, chest pain, weight loss, night sweats and fever. With the exception of the following; pregnant or possibly pregnant women, those who for one reason or the other could not take or declined taking the chest X-ray, everyone's X-ray was taken using the digital X-ray machine. A physician assistant trained to read digital X-ray images categorized the X-ray finding into 3 groups: normal, abnormal suggestive of TB and abnormal but unlikely to be TB. Those reported to have an abnormal X-ray suggestive of TB, those admitting having a cough of 2 weeks or any duration with at least one other symptom were presumed to have TB and were requested to produce 1 spot sample of sputum. Also included in the presumed TB group and asked to produce sputum

for testing were those unable to have an X-ray taken as indicated above and HIV positive individuals. The samples were transported to designated laboratories with a GeneXpert machine for testing. The GeneXpert results were reported as Mycobacterium tuberculosis (MTB) not detected, MTB detected rifampicin sensitive and MTB detected rifampicin resistance detected. The field coordinator followed up on the results and arranged for those found to have TB to be registered at the district level to start TB treatment. The analyses of the treatment outcomes of these TB patients were beyond the scope of this paper.

**Statistical analyses.** The NTP dataset contained data on only the people who volunteered for the screening and did not have data on the total number of people in the communities nor the demographics of the total community population. The analyses for the study was therefore limited to the population that was screened. Descriptive analyses were conducted for the following variables district, age group, sex, occupation, X-ray reading, presence of TB symptoms and GeneXpert results according to region. Due to the lack of census data of the communities targeted, screening coverage, (the proportion of the population screened) could not be assessed; neither was it possible to assess how the study population (the people who showed up for the screening) compared to the demographics of the communities from which they came. Data on the proportions of age groups and sex by region of persons 15 years and above was however obtained from the 2010 Ghana census data and used as proxy data for the communities to allow for comparison of age group and sex with the study population with the assumption that the community demographics were similar to the census data [17]. Among the population screened, the prevalence of TB was calculated and so were the number needed to screen (NNS) and the number needed to test (NNT) to detect a TB case.

Univariate logistic regression was initially conducted to assess factors associated with TB diagnosis. Given the few characteristics being examined, all variables were then entered into a binomial logistic regression model with the outcome variable being a diagnosis of TB confirmed by GeneXpert. Odds ratio and 95% confidence (95%CI) interval were assessed. The strength of association between symptoms and having X-ray suggestive of TB was also determined by assessing Pearson's phi correlation coefficient as well as multivariable logistics regression controlling for age and sex. Finally, the diagnostic value of the screening methods chest X-ray and symptoms in the detection of TB cases was also evaluated using receiver operating characteristic (ROC) curves. The predictive strengths were determined from the models by calculating the area under the ROC curve (AUC). In the sensitivity analysis, all those not tested for TB were assumed not to have TB. STATA version 13 was used for statistical analyses and p<0.05 was set as the level of significance. The Ghana Health Service Ethical Review Committee gave ethical approval for the conduct of the study.

## Results

### Demographic characteristics and yield of TB among screened population

From among the 3 regions Brong Ahafo, Ashanti and Western, a total of 10,441 people from 78 communities in 21 districts were screened. On the average, 226 people were screened a day. The number of people screened per community ranged from 39 to 202 while for the districts, the numbers screened ranged from 282 to 826. The demographic characteristics of the study population by region and the census data are presented in Table 1. Overall among those screened, 45% were male compared to 49% in the census data showing that among those screened, slightly fewer men showed up. While census data showed that among those aged 15 years and above in the general population about 33% were between 15 to 24 years, among those screened about 12% were in this age group. Those screened in the TB screening exercise consisted of a relatively higher proportion of older people compared to what would be

**Table 1. Demographic characteristics and TB screening results of person screened in mining communities in Ashanti, Brong Ahafo and Western Regions, Ghana, May 2017 to January 2018 compared to 2010 regional census statistics.**

| | Ashanti Total = 1896 | Proportions of age groups 15+ from Ashanti census data | Brong Ahafo Total = 3252 n (%) | Proportions of age groups 15+ from Brong Ahafo census data | Western Total = 5293 n (%) | Proportions of age groups 15+ from Western census data |
|---|---|---|---|---|---|---|
| **Districts** | 3 | | 6 | | 12 | |
| **Communities** | 14 | | 24 | | 40 | |
| | n (%) | (%) | n (%) | (%) | n (%) | (%) |
| **Age (years)** | | | | | | |
| <15 | 0 | | 13 (0.4) | | 10 (0.2) | |
| 15–24 | 240 (12.7) | 33.2 | 357 (11.0) | 33.9 | 686 (13.0) | 33.2 |
| 25–34 | 368 (19.4) | 25.2 | 676 (20.8) | 24.0 | 1,225 (23.1) | 24.9 |
| 35–44 | 351 (18.5) | 17.3 | 653 (20.1) | 17.0 | 1,095 (20.7) | 17.9 |
| 45–54 | 350 (18.5) | 11.3 | 581 (17.9) | 11.5 | 997 (18.8) | 11.8 |
| 55–64 | 312 (16.5) | 6.1 | 433 (13.3) | 6.1 | 650 (12.3) | 6.1 |
| 65+ | 272 (14.3) | 6.9 | 536 (16.5) | 7.5 | 620 (11.7) | 6.2 |
| Missing | 3 (0.2) | | 3 (0.1) | | 9 (0.2) | |
| **Sex** | | | | | | |
| Male | 882 (47.0) | 48.4 | 1481 (45.5) | 49.6 | 2,268 (42.9) | 50.0 |
| Female | 995 (52.5) | 51.6 | 1769 (54.4) | 50.4 | 3,018 (57.0) | 50.0 |
| Missing | 9 (0.5) | | 2 (0.1) | | 7 (0.1) | |
| **Occupation** | | | | | | |
| Agriculture | 771 (37.5) | | 1639 (50.4) | | 1,911 (36.1) | |
| Professional | 130 (6.9) | | 242 (7.4) | | 250 (4.7) | |
| Miner | 103 (5.4) | | 278 (8.6) | | 486 (9.2) | |
| Sales and services | 497 (26.2) | | 587 (18.1) | | 1,496 (28.3) | |
| Manual labour | 114 (6.0) | | 281 (8.6) | | 447 (8.5) | |
| Unemployed | 319 (16.8) | | 182 (5.6) | | 489 (9.2) | |
| Missing | 22 (1.2) | | 43 (1.3) | | 214 (4.0) | |
| **Symptoms** | | | | | | |
| Cough any duration | 484 (25.5) | | 575 (17.7) | | 1,074 (20.3) | |
| Sputum | 2 (0.1) | | 232 (7.1) | | 50 (0.9) | |
| Chest pain | 527 (27.8) | | 336 (10.3) | | 772 (14.6) | |
| Weight loss | 442 (23.3) | | 154 (4.7) | | 644 (12.2) | |
| Night sweats | 309 (16.3) | | 206 (6.3) | | 518 (9.8) | |
| Fever | 251 (13.2) | | 57 (1.8) | | 399 (7.5) | |
| No symptoms | 1037 (54.7) | | 2388 (73.4) | | 3697 (69.8) | |
| **X-ray Results** | | | | | | |
| Normal | 1,082 (57.1) | | 2937 (90.3) | | 3,572 (67.5) | |
| Abnormal TB | 387 (20.4) | | 213 (6.6) | | 848 (16.0) | |
| Abnormal Non-TB | 413 (21.8) | | 79 (2.4) | | 798 (15.1) | |
| Missing | 14 (0.7) | | 23 (0.7) | | 75 (1.4) | |

expected in the community using census data as proxy. This general trend was noted across the 3 regions. There were 23 children (0.2%) less than 15 years. About 42% of the people screened were in the agriculture sector while miners comprised 8.5%.

About half of those screened 50.7%, were from Western Region. Ashanti Region had the smallest proportion of miners 5.4% while 50.4% of those screened in Brong Ahafo were in the

agriculture sector. The districts in which the TB case finding activities were undertaken are shown in Fig 1.

Fig 2 shows the algorithm of the overall TB screening process and the yield.

Among those who had a chest X-ray done (10,329), almost 14% (1,448) were reported to have findings suggestive of TB. About 23% of those screened were identified as presumed TB cases. The 95 TB cases identified out of the 10,441 people screened was equivalent to 910 TB cases per 100,000 population screened. Out of the 78 communities, thirty-seven (47%) did not have any TB cases detected, fourteen communities had 1 TB case, sixteen had 2, two had 3 TB cases detected. Six communities had 4 TB cases, two had 5 and 1 had 9 TB cases detected. The TB prevalence among the population screened varied between 0 and 6.8%. No TB cases were identified in 3 districts out of the 21 districts undertaking TB case finding activities. Ashanti recorded the highest yield of TB cases among those screened 1.32%, compared to Western 1.11% and Brong Ahafo 0.34%. Table 2 shows the yield of TB cases by regions following TB case finding exercises undertaken in selected mining communities in selected districts

Western Region accounted for 62% (59) of the TB cases while 26% (25) were from Ashanti Region. Males made up 84% (80) of the total TB cases detected among the population screened, 70% of those with TB were in the age range of 25 to 54 years and miners had the highest prevalence of TB 2.65% compared to 1.41% among those engaged in skilled manual labour and 0.68% among those involved in the agriculture sector. The overall TB prevalence among non-miners in the screened population was 0.64%. Five of the TB cases had rifampicin resistant TB. All 5 were male in the 35 to 44 year age group with 2 of them being miners from Western Region giving a rate of 8.7% rifampicin resistant TB among the miners with TB.

The regional distribution of TB cases across the demographic variables is shown in Table 3.

In the Ashanti Region, all but 1 of the 25 TB case identified were male. Even though miners formed 9.6% of those screened in the Western Region, 27% of those with TB in the region were miners.

## Factors associated with TB

In univariate analyses, shown in Table 4, being a male, miner and unemployed were risk factors for having TB as was coming from Western and Ashanti regions. In multivariate analyses also in Table 4, being in the age group 35 to 54 years became a risk factor for TB, (OR [aOR] 2.27, 95% CI 1.35–3.84, P = 0.002). Being male, (OR [aOR] 5.96, 95% CI 3.25–10.92, P < 0.001) miners (OR [aOR] 2.70, 95% CI 1.47–4.96, P = 0.001), unemployed (OR [aOR] 3.15, 95% CI 1.66–5.95, P < 0.001) and coming from Ashanti (OR [aOR] 4.12, 95% CI 1.89–8.98, P < 0.001) and Western Regions (OR [aOR] 3.84, 95% CI 1.90–7.80, P < 0.001) were risk factors for having TB.

*Correlation between TB symptoms and X-ray suggestive of TB*. The analysis to assess the correlation between the symptoms and X-ray suggestive of TB is shown in Table 5. Cough of any duration was found to have the strongest correlation with X-ray suggestive of TB with a correlation coefficient of 0.48 pointing to a strong positive relationship while sputum ranked at the bottom.

In the analysis assessing the performance of symptoms and X-ray at screening and a diagnosis of TB confirmed with GeneXpert, cough was also the most sensitive symptom for a diagnosis of TB at a sensitivity of 86.3% (95% CI 79.4–93.2) followed by weight loss 67.4% (95% CI 57.0–76.6) and chest pain 63.2% (95% CI 52.6–72.8) with sensitivities that were overlapping Table 6.

Sputum production had the least sensitivity 8.4% (95% CI 3.7–15.9). The sensitivity of X-ray in the diagnosis of TB 81.1% (95% CI 71.7–88.4) ranked after that of cough but was

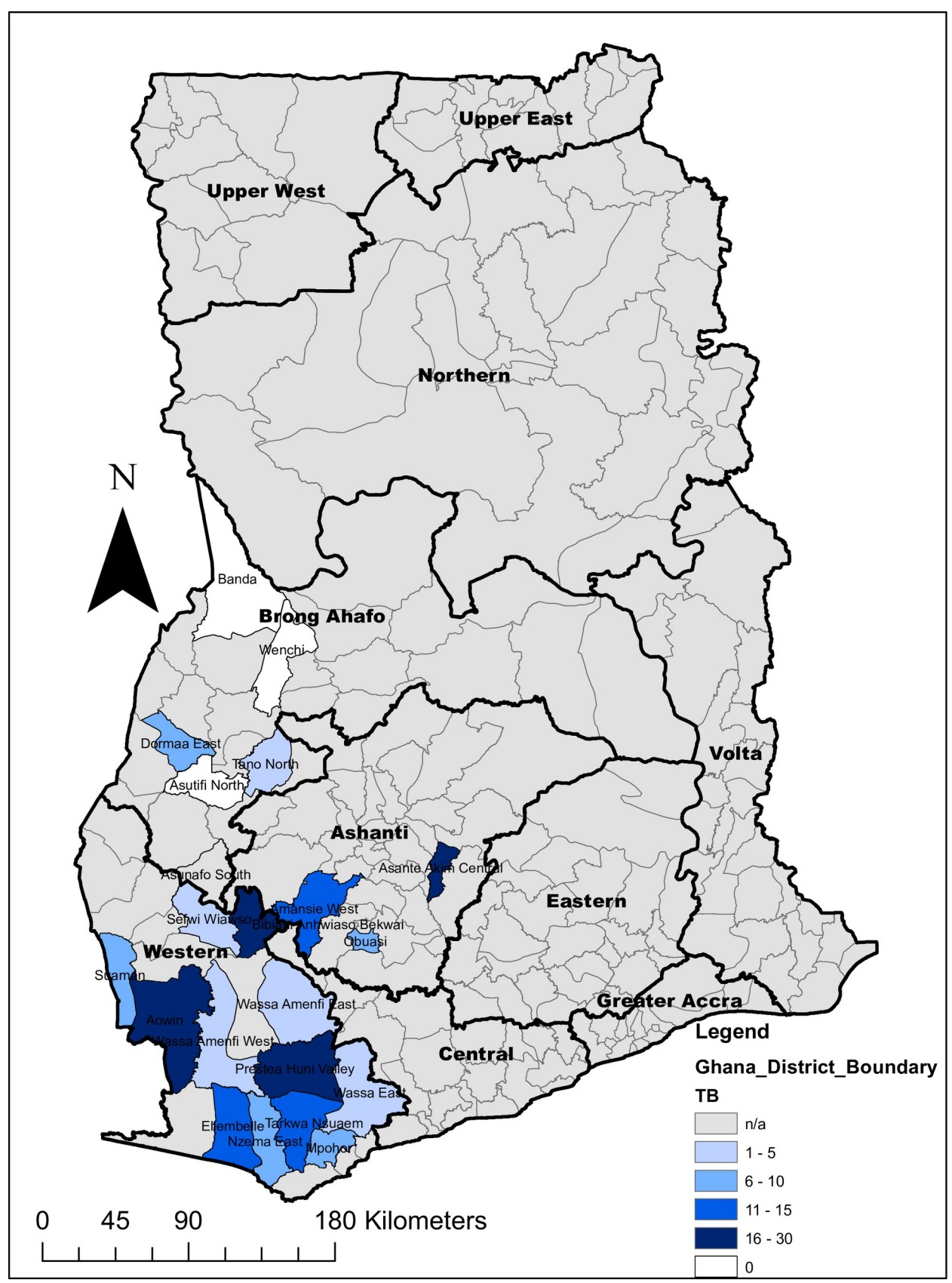

**Fig 1. Map of Ghana showing districts in which case finding activities were undertaken in selected mining communities from May 2017 to January 2018.** Fig 1 legend shows the range of the number of TB cases detected per 1000 persons screened from the TB case finding exercise in the respective districts.

comparable. The combination of cough and chest X-ray had the highest sensitivity 94.7 (95% CI). The specificities of the symptoms and X-rays ranged from 80.2% (cough) to 97.3% (sputum). The positive predictive values of all the symptoms and X-ray suggestive of TB were all low <7%. The diagnostic value of the screening methods, summarized by the AUC, highlighted cough (0.83) and X-ray (0.84) as being at par.

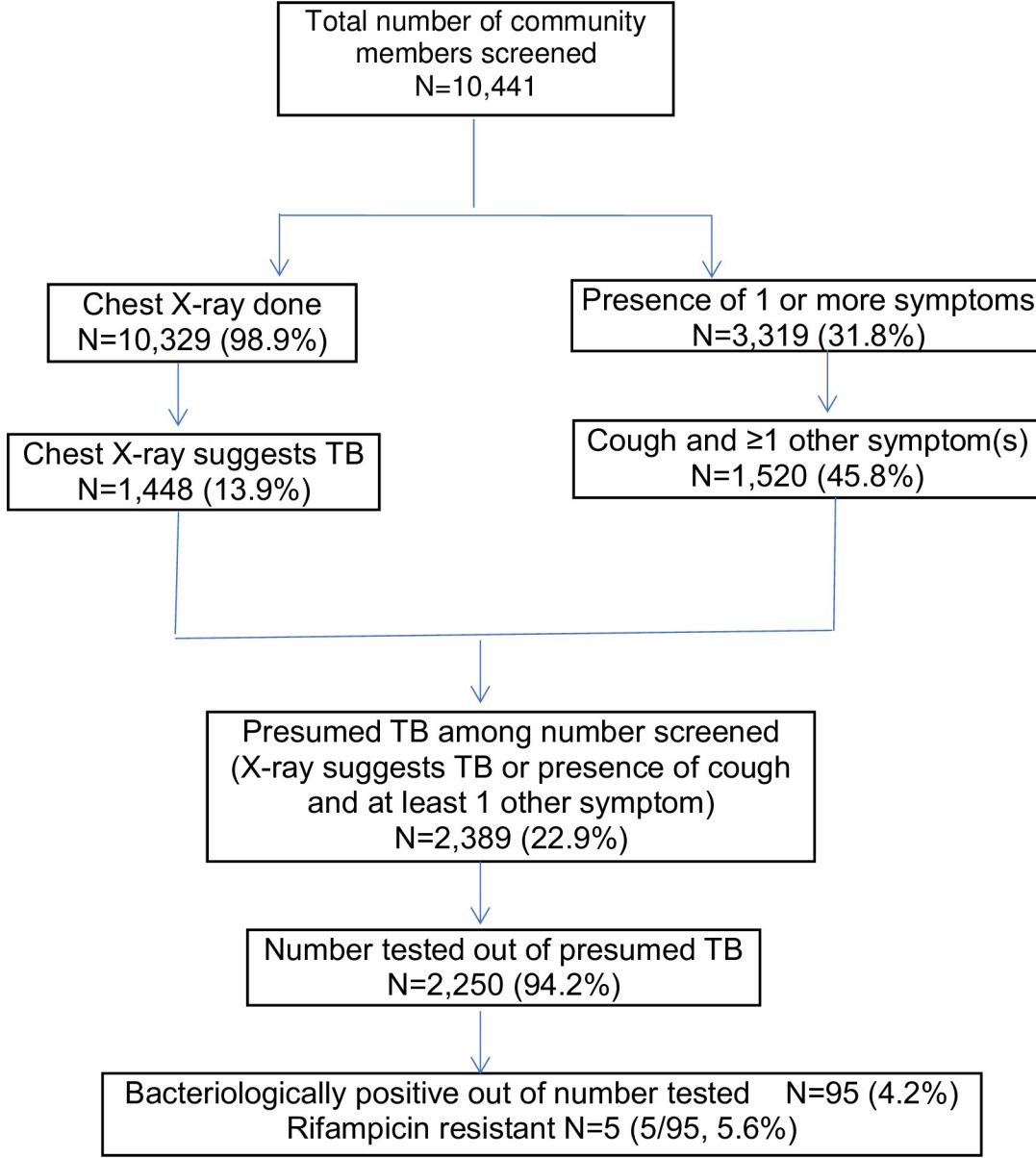

**Fig 2. Flow chart of TB case finding screening exercise undertaken in mining communities in 21 districts in Ghana from May 2017 to January 2018.**

**Table 2. Yield of TB cases from TB case finding exercise undertaken in 3 regions in Ghana, May 2017 to January 2018.**

| Regions | Ashanti | Brong Ahafo | Western | Total |
|---|---|---|---|---|
| Number screened | 1896 | 3252 | 5293 | 10441 |
| Presumptive TB (% out of screened) | 444 (23.4) | 536 (16.5) | 1,409 (26.0) | 2,389 (22.9) |
| Number tested (% of presumptive TB) | 441 (99.3) | 401 (74.8) | 1408 (99.9) | 2250 (94.2) |
| Number bacteriologically positive (% out of tested) | 25 (5.6%) | 11 (2.8%) | 59 (4.2) | 95 (4.2) |
| Number rifampicin resistant (% out of TB cases) | 1 (4%) | 1 (9.1%) | 3 (5.1) | 5 (5.3) |
| NNT (Number needed to test) | 18 | 36 | 24 | 24 |
| NNS (Number needed to screen) | 76 | 296 | 90 | 110 |

## Discussion

Miners and the communities in which they live are identified as being at higher risk for TB than the general population [18]. Forging ahead with the End TB Strategy, responsive programs for such key populations require data that will inform and guide stakeholders in the appropriate planning and delivering of services [4,19]. In this study, which is the first to the best of our knowledge to highlight active TB case finding among artisanal mining communities in Ghana using mobile X-ray and symptom screening, the study population consisted of people who volunteered for the screening exercise. The overall rate of TB among the people screened was found to be 910 per 100,000 people screened with 5.3% of the TB cases having rifampicin resistant TB. The NNS and NNT were 110 and 24 respectively. Risk factors for TB were being male, a miner and location in Ashanti and Western Regions. Cough regardless of

**Table 3. Distribution of TB and those without TB across demographic variables among persons screened in selected artisanal mining communities by region in Ghana May 2017 to January 2018.**

| | Ashanti N = 1896 | | Brong Ahafo N = 3252 | | Western N = 5293 | |
|---|---|---|---|---|---|---|
| | No TB | TB | No TB | TB | No TB | TB |
| | N = 1871 | N = 25 | N = 3241 | N = 11 | N = 5234 | N = 59 |
| **Age Years)** | | | | | | |
| <15 | 0 | 0 | 13 (0.4) | 0 | 10 (0.2) | 0 |
| 15–24 | 240 (12.8) | 0 | 357 (11.0) | 0 | 681 (13.0) | 5 (8.5) |
| 25–34 | 361 (19.3) | 7 (28.0) | 674 (20.8) | 2 (18.2) | 1212 (23.2) | 13 (22.0) |
| 35–44 | 345 (18.4) | 6 (24.0) | 650 (20.1) | 3 (27.3) | 1079 (20.6) | 16 (27.1) |
| 45–54 | 346(18.5) | 4 (16.0) | 579 (17.9) | 2 (18.2) | 983 (18.8) | 14 (23.7) |
| 55–64 | 307(16.4) | 5 (20.0) | 432 (13.3) | 1 (9.1) | 642 (12.3) | 8 (13.6) |
| 65+ | 269 (14.4) | 0 | 533 (16.4) | 3 (27.3) | 618 (11.8) | 3 (5.1) |
| Missing | 3 (0.2) | | 3 (0.1) | 0 | 9 (0.2) | 0 |
| **Sex** | | | | | | |
| Male | 868 (46.4) | 24 (96.0) | 1472 (45.4) | 9 (82.8) | 2221 (42.4) | 47 (79.7) |
| Female | 994 (53.1) | 1 (4.0) | 1767 (54.5) | 2 (18.2) | 3006 (57.4) | 12 (20.3) |
| Missing | 9 (0.5) | 0 | 2 (0.1) | 0 | 7 (0.1) | 0 |
| **Occupation** | | | | | | |
| Agriculture | 700 (37.4) | 11 (44.0) | 1636 (50.5) | 3 (27.3) | 1896 (36.2) | 15 (25.4) |
| Professional | 130 (7.0) | 0 | 242 (7.5) | 0 | 250 (0.8) | 0 |
| Miner | 98 (5.0) | 5 (20.0) | 276 (8.5) | 2 (18.2) | 470 (9.0) | 16 (27.1) |
| Sales and services | 495 (26.5) | 2 (8.0) | 586 (18.1) | 1 (9.1) | 1487 (28.4) | 9 (15.3) |
| Manual lab | 112 (6.0) | 2 (8.0) | 278 (8.6) | 3 (27.3) | 442 (8.4) | 5 (8.5) |
| Unemployed | 314 (16.8) | 5 (20.0) | 182 (5.6) | 0 | 476 (9.1) | 13 (22.0) |
| Missing | 22 (1.2) | 0 | 41 (1.3) | 2 (18.2) | 213 (4.1) | 1 (1.7) |

**Table 4. Logistic regression analyses of factors associated with TB diagnosis among persons screened in selected artisanal mining communities in Ghana, May 2017 to January 2018, N = 10,441.**

| | Univariate OR (%CI) | p | Multivariate aOR (95CI) | p |
|---|---|---|---|---|
| **Age (years)** | | | | |
| <35 | 1 | | 1 | |
| 35–54 | 1.49 (0.92–2.40) | 0.106 | 2.27 (1.35–3.84) | 0.002 |
| ≥55 | 1.08 (0.62–1.89) | 0.789 | 1.82 (0.98–3.39) | 0.055 |
| **Sex** | | | | |
| Female | 1 | | 1 | |
| Male | 6.74 (3.88–11.72) | <0.001 | 5.96 (3.25–10.92) | <0.001 |
| **Occupation** | | | | |
| Agriculture | 1 | | 1 | |
| Professional | - | | - | |
| Miner | 3.98 (2.29–6.91) | <0.001 | 2.70 (1.47–4.96) | 0.001 |
| Sales and services | 0.68 (0.35–1.34) | 0.266 | 1.12 (0.55–2.29) | 0.751 |
| Manual labour | 1.75 (0.85–3.61) | 0.127 | 1.08 (0.51–2.30) | 0.738 |
| Unemployed | 2.70 (1.49–4.89) | 0.001 | 3.15 (1.66–5.95) | <0.001 |
| **Region** | | | | |
| Brong Ahafo | 1 | | 1 | 1 |
| Ashanti | 3.94 (1.93–8.02) | <0.001 | 4.12 (1.89–8.98) | <0.001 |
| Western | 3.32 (1.74–6.33) | <0.001 | 3.84 (1.90–7.80) | <0.001 |

OR–Odds ratio, aOR–adjusted odds ratio.

**Table 5. Correlation between symptoms among persons screened in selected artisanal mining communities in Ghana, May 2017 to January 2018 and X-ray suggestive of TB using Phi Correlation Coefficient.**

| | X-ray suggestive of TB N = 1,448 | X-ray not suggestive of TB N = 8,881 | Phi correlation coefficient (r) | p |
|---|---|---|---|---|
| **Cough any duration** | | | | |
| Yes | 988 (68.2) | 1,122 (12.6) | | |
| No | 460 (31.8) | 7,759 (87.4) | 0. 48 | <0.001 |
| **Sputum** | | | | |
| Yes | 146 (10.1) | 135 (1.5) | | |
| No | 1,302 (89.9) | 8,746 (98.5) | 0.18 | <0.001 |
| **Chest pain** | | | | |
| Yes | 686 (47.4) | 939 (10.6) | | |
| No | 762 (52.6) | 7,942 (89.4) | 0.35 | <0.001 |
| **Weight loss** | | | | |
| Yes | 612 (42.3) | 625 (7.0) | | |
| No | 836 (57.7) | 8,256 (93.0) | 0.38 | <0.001 |
| **Night sweats** | | | | |
| Yes | 488 (33.7) | 541 (6.1) | | |
| No | 960 (66.3) | 8,340 (93.9) | 0.32 | <0.001 |
| **Fever** | | | | |
| Yes | 372 (25.7) | 334 (3.8) | | |
| No | 1,076 (74.3) | 8,547 (96.2) | 0.30 | <0.001 |

95%CI– 95% Confidence Interval, aOR Adjusted Odds Ratio.

**Table 6. Sensitivity and specificity of symptoms and X-ray at screening in the diagnosis of TB.**

| | TB cases | No TB | Sensitivity (95%CI) | Specificity (95%CI) | PPV (95%CI) | AUC |
|---|---|---|---|---|---|---|
| **Cough any duration** | | | | | | |
| Yes | 82 | 2,051 | 86.3 (77.7–92.5) | 80.2 (79.4–80.9) | 3.8 (3.1–4.8) | 0.83 (0.80–0.87) |
| No | 13 | 8,295 | | | | |
| **Sputum** | | | | | | |
| Yes | 8 | 276 | 8.4 (3.7–15.9) | 97.3 (97.0–97.6) | 2.8 (1.2–5.5) | 0.53 (0.50–0.56) |
| No | 87 | 10,070 | | | | |
| **Chest pain** | | | | | | |
| Yes | 60 | 1,575 | 63.2 (52.6–72.8) | 84.8 (84.1–85.5) | 3.7 (2.8–4.7) | 0.74 (0.69–0.79) |
| No | 35 | 8,771 | | | | |
| **Weight loss** | | | | | | |
| Yes | 64 | 1,176 | 67.4 (57.0–76.6) | 88.6 (88.0–89.2) | 5.2 (4.0–6.5) | 0.78 (0.73–0.83) |
| No | 31 | 9,170 | | | | |
| **Night sweats** | | | | | | |
| Yes | 34 | 999 | 35.8 (26.2–46.3) | 90.3 (89.8–90.9) | 3.3 (2.3–4.6) | 0.63 (0.58–0.68) |
| No | 61 | 9,347 | | | | |
| **Fever** | | | | | | |
| Yes | 35 | 672 | 36.8 (27.2–47.4) | 93.5 (93.0–94.0) | 4.9 (3.5–6.8) | 0.65 (0.60–0.70) |
| No | 60 | 9,674 | | | | |
| **X-ray** | | | | | | |
| Yes | 77 | 1,371 | 81.1 (71.7–88.4) | 86.6 (85.9–87.3) | 5.3 (4.2–6.5) | 0.84 (0.80–0.88) |
| No | 18 | 8,863 | | | | |
| **Cough and X-ray** | | | | | | |
| Yes | 90 | 2,503 | 94.7 (88.1–98.3) | 75.8 (75.0–76.6) | 3.5 (2.8–4.3) | 0.85 (0.83–0.88) |
| No | 5 | 7,843 | | | | |

95%CI– 95% Confidence Interval, PPV–Positive predictive value, AUC–Area under the curve.

duration correlated with X-ray findings suggestive of TB and a diagnosis of TB. It is not straightforward to compare these findings from our study population (in which sampling was not done) to others from TB prevalence and similar studies in which the populations are sampled and the results reasonably reflect the communities in which the surveys are conducted. Given the different methodologies, contexts and the point that it was not possible to tell whether the people screened in our study fairly represented the rest of the community, interpretations of these comparisons should therefore consider the background of our study setting.

Using census data as proxy for the community demographics, it was noted that fewer men and relatively older community members offered themselves for the screening exercise in our study. Some studies conducted among small scale miners in Ghana show that about 70% or more of those surveyed were less than 40 years suggesting one would expect more younger men in such mining communities [20,21]. It may therefore be possible that the screening exercise conducted in the AMC in our study missed relatively younger people. In case, a significant proportion of those missed were miners, given the higher risk of TB among miners, it could be possible that our results may be an underestimation of the TB burden in AMC studied [18]. The prevalence of TB among the people screened in the artisanal mining communities in this study was more than two and half times what was reported for the general population in Ghana in the TB prevalence survey (356 bacteriologically confirmed TB per 100,000), on the face value re-iterating the data on the higher risk of TB among mining communities [9,18,22].

The overall TB prevalence among the community members screened was however lower than that reported in other mining communities such as the copper belt in Zambia (1.2%) and mining communities in Myanmar (2.7%) [11,23]. The TB prevalence among the miners in our study (2.65%), while also lower than that cited in other studies in South Africa (5.4%) and Zambia (9.5%), was higher than that in miners in Myanmar (1.7%) [11,23,24]. These differences may be related to several factors including varying methods for diagnosing TB in the miners such as sputum microscopy as opposed to GeneXpert, the type of mining, the exposure to silica dust and the duration of exposure to mining conditions [23]. In the Zambia study, study participants were underground miners in copper mines while the miners in our study were likely involved in gold exploration using different mining methods ranging from surface to underground mining [15,16,23,25,26].

The rate of rifampicin resistance found among our study population (5.3%) was slightly higher than the 4.3% drug resistance recorded among South African gold miners and their dependents but much higher than the 1.3% reported from the 2018 national surveillance data from Ghana [27,28]. The setting in the South Africa study was a formal gold mine employment with a well-functioning TB control program that was keeping track of drug resistant TB. In contrast, but for the outreach program to these mining communities, it is not possible to tell how long it would have taken for these drug resistant cases to be detected. This highlights the importance of mainstreaming TB screening services for such high-risk populations [29].

In general males are more at risk for TB in Ghana and other countries and our study findings are in line with this [22,29]. It is reported that in illegal artisanal small-scale mining settings in Ghana, women may constitute up to 50% of the labour force and play various roles [16,30,31]. The men however undertake more hazardous activities including underground mining which may expose them more to silica increasing the risk for TB [23,30–32].

TB prevalence surveys have shown chest X-rays to be more sensitive than symptom screening for TB detection since some people with TB may not have symptoms [33]. In our study the sensitivity and specificity of chest x-ray as a screening tool for TB fell within the range of what has been reported [34]. Symptom screening, particularly cough of any duration, on the other hand is reported to be less sensitive with pooled sensitivity and specificity in the range of 40 to 74% and 69 to 90% respectively [34]. Several studies on the diagnostic value of x-ray and symptoms screening for TB case detection have been conducted in different populations and among miners but there's a dearth of such studies in artisanal mining communities to compare with [5,35–39]. Comparing across such studies may however not be straightforward because of the different modalities of symptom definitions, screening criteria and subjectivity in reading of X-rays [38,40]. In our study the sensitivity (86.3%) of cough of any duration as a screening tool for the detection of TB was relatively high and even ranked higher than that of chest X-ray even though it was comparable. The combination of cough and X-ray raised the sensitivity to 95%. This finding is similar to what van't Hoog and colleagues reported in a prevalence study in Kenya even though the cough in their study was 2 weeks or more in duration [38].

In this TB case detection outreach to the artisanal mining communities, those who were eligible to produce sputum for testing consisted of those who had an X-ray suggestive of TB, symptoms suggestive of TB and those who for some reason could not take X-ray totalling about 23% of the population who presented themselves for screening. This is relatively higher compared to the 13% identified during the TB prevalence survey [17]. The difference may be related to the expertise in reading the X-rays (radiologists in the prevalence survey as opposed to physician assistants in our survey), the cough criteria (2 weeks cough in the prevalence survey) the possibility that some of those who had symptoms or were reported as having X-ray suggestive of TB could also have had silicosis which may have symptoms and X-ray findings similar to that of TB [41,42]. It is also however important to note that, in the TB prevalence

survey the population screened were sampled unlike the study population in our study who came voluntarily for screening with the possibility of some selection bias. The yield of TB cases was however much higher and may make the case for the use of this approach of combining symptom and x-ray screening used in the mining communities as opposed to a sequential algorithm of symptom screening following by X-ray [33]. Other studies have explored sequential algorithms in active TB case finding but in settings different from our study which limit comparability with our results [40,43]. The commonality however lies in the benefit of including X-ray to enhance the identification of TB in the case finding exercise. Further studies assessing a cost benefit analyses and factoring in yield of TB cases may shed more light on the more beneficial and cost-effective approach to use in high risk populations in hard to reach communities.

This study has a number of limitations. The NTP dataset used for the analysis did not have data on the number of people in the communities in which screening was conducted nor the demographics of the communities. This limited the ability to assess the size of the community, the proportion of the community population that was screened and whether the demographics of those screened were comparable to those not screened. The diagnosis of TB was based on results from GeneXpert to the exclusion of those who may have had clinically diagnosed TB. Having used routine data already collected, there was no control over standardizing the measurement and reporting of variables to enable further analyses of interest nor was it not possible to apply quality control measures or checks. For example, in the data set the recording cough of any duration without always specifying the duration for each participant limited the ability to assess the sensitivities of the different durations of cough in the diagnosis of TB. It is therefore being recommended to the NTP to have standardized data collection tools for all teams engaged in outreach programs to enable optimum analyses of data to inform programming. Finally, with this study being conducted in artisanal mining communities the findings may not be generalizable to other communities in the country.

## Conclusion

The high risk of TB in the artisanal mining communities and in miners as shown in this study reinforces the need to target these populations with outreach programs particularly given that they may be in hard to reach areas. Even though the combination of cough and chest X-ray had the highest sensitivity, the diagnostic value of cough highlights the usefulness of symptom screening in this population that may be harnessed even in the absence of X-ray to identify those suspected to have TB for TB diagnostic evaluation.

## Supporting information

**S1 File. Data set for TB case finding activities.**
(XLSX)

## Acknowledgments

The authors acknowledge with much appreciation the staff of the Ghana National Tuberculosis Control Program, regional health directorates of Ashanti, Brong Ahafo and Western Regions, the staff of the district health directorates of the participating districts, the communities that were screened and the various screening teams for the diverse roles played in the TB case finding exercise. Their efforts enabled the generation of the data for this study. Dr Rafiq Okine, Mr Fred Osei-Sarpong and Mr Felix Sorvor are also appreciated for the technical support.

## Author Contributions

**Conceptualization:** Sally-Ann Ohene, Frank Bonsu, Yaw Adusi-Poku, Francisca Dzata, Mirjam Bakker.

**Data curation:** Sally-Ann Ohene, Francisca Dzata.

**Formal analysis:** Sally-Ann Ohene, Mirjam Bakker.

**Methodology:** Sally-Ann Ohene, Mirjam Bakker.

**Supervision:** Mirjam Bakker.

**Writing – original draft:** Sally-Ann Ohene.

**Writing – review & editing:** Sally-Ann Ohene, Frank Bonsu, Yaw Adusi-Poku, Francisca Dzata, Mirjam Bakker.

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
