## [Decision Letter · Decision Letter 0]

18 Jan 2021

PONE-D-20-34085

Case finding of tuberculosis among mining communities in Ghana

PLOS ONE

Dear Dr. Ohene,

Thank you for submitting your manuscript to PLOS ONE. After careful consideration, we feel that it has merit but does not fully meet PLOS ONE’s publication criteria as it currently stands. Therefore, we invite you to submit a revised version of the manuscript that addresses the points raised during the review process.

Please submit your revised manuscript. If you will need significantly more time than this to complete your revisions, please reply to this message or contact the journal office at plosone@plos.org. Please include the following items when submitting your revised manuscript:

We look forward to receiving your revised manuscript.

Kind regards,

Frederick Quinn

Academic Editor

PLOS ONE

Journal Requirements:

2. Please state whether the data utilized in this study were de-identified/anonymised before access?

3. We note that Figure 1 in your submission contains map images which may be copyrighted.

We require you to either (a) present written permission from the copyright holder to publish this figure specifically under the CC BY 4.0 license, or (b) remove the figure from your submission:

b. If you are unable to obtain permission from the original copyright holder to publish this figure under the CC BY 4.0 license or if the copyright holder’s requirements are incompatible with the CC BY 4.0 license, please either i) remove the figure or ii) supply a replacement figure that complies with the CC BY 4.0 license. Please check copyright information on all replacement figures and update the figure caption with source information. If applicable, please specify in the figure caption text when a figure is similar but not identical to the original image and is therefore for illustrative purposes only.

Reviewers' comments:

Reviewer's Responses to Questions

**Comments to the Author**

1. Is the manuscript technically sound, and do the data support the conclusions?

Reviewer #1: Partly

Reviewer #2: Yes

2. Has the statistical analysis been performed appropriately and rigorously? 

Reviewer #1: No

Reviewer #2: Yes

3. Have the authors made all data underlying the findings in their manuscript fully available?

Reviewer #1: Yes

Reviewer #2: Yes

4. Is the manuscript presented in an intelligible fashion and written in standard English?

Reviewer #1: Yes

Reviewer #2: Yes

5. Review Comments to the Author

Reviewer #1: The study by Ohene et al. describe active case-finding (ACF) for TB in several mining communities in Ghana and presents novel data that have potentially wider relevance to ACF in other countries, including what types of screening should be performed and what kind of yields can be expected. The authors should also be commended for undertaking a study based on routine data collected by the national TB program which remains an underutilized resource. However, the manuscript has a number of significant issues that need to be resolved.

The first primary issue with the manuscript is that it fails to describe completely how the screening was performed in each of the 3 regions, 21 districts, and 78 communities included in the study. Without these additional data, it is difficult to know what the true denominator for the study was and how representatively the communities were sampled. In particular, I was looking for additional details in the Results (lines 192-217) such as:

- What was the estimated catchment size of the outreach performed in each community?

- What kind of catchment was expected as a result of the community outreach and radio announcements? (For example, in the South Africa Thibela TB study, Grant et al. described these activities in a separate paper, AIDS 2010 24:S37.)

- What were the demographics of these communities compared to the demographics of the study population (i.e., the people who showed up for screening)?

- In addition, Table 1 presenting the demographics of the persons screened is difficult to interpret without the demographics of the regions to compare them against. These data ought to be available from census or modeled projects such as WorldPop.

- Are there reasons to believe the sample was biased in any way? For example, did fewer men or mining workers show up than expected? If there was bias, was the bias different for each region, district, or community?

- The authors state that the people screened per community "ranged from 39 to 202" (line 194). Are these meaningful differences in terms of the community population sizes?

As a result of the above exclusions, I find some of the claims and intended comparisons unsupported. For example, in several instances, the authors discuss the prevalence in the communities. For example, in the Results the authors state "the community prevalence varied between 0 and 6.8%" (line 232). Later, in the Discussion the authors compare these results on "the overall community TB prevalence" to published results in other countries such as South Africa and Zambia (lines 335-340). However, without a proper accounting of the sampling and any correction between the study population and the overall target population, these claims are unjustified. (For example, one can imagine a scenario where only the most ill mine workers showed up in one region with good outreach, while less ill mine workers and perhaps their families and friends showed up in another region.)

The authors should exercise care and qualify the language around these claims, in addition to explaining how screening was performed and what the demographics of the communities were. The authors make a partial acknowledgment late in the Discussion ("Comparing across such studies may however not be straightforward..." lines 368). However, by then the claims have already been made.

The second primary issue involves the statistics in the manuscript, some of which are either insufficiently described or perhaps even a bit suspect. These statistics and their accompanying tables should be revised for clarity and checked for rigor.

- In Table 3, describing the bivariate analyses, it is not clear what statistical test was performed to arrive at a single p-value per demographic variable per region. What test is this? What is the null hypothesis in each case? Why were the 3 regions analyzed separately rather than jointly in a hierarchical/mixed effects model? Also, claims in this section do not appear supported by the results. For example, "Those with TB were more likely to be male" in line 261. What test and test statistic supports this?

- Table 4 presents the results of a logistic regression, but again here, the model used is unclear. The authors should state the logistic regression equation model in the Methods. In addition, the accompanying text should clarify the difference in purpose between Table 3 and Table 4, as they appear to have the same function.

- Table 5 is unclear in several aspects. What does "X-ray suggestive of TB" mean? Is this the same as, say, an X-ray with Xpert-confirmed TB? What is the purpose of this section (from line 278)? If it is to establish that particular symptoms may be as good as Xpert-confirmed X-ray screening, the authors should state so. Also, how should the reader interpret the kappa statistic, compared to say the more familiar correlation coefficient?

Finally, the authors bring up a good point at the end of the Discussion that should be raised earlier, comparing combined symptom and X-ray screening compared to a sequential algorithm. In particular, the former seems faster and perhaps easier, and it would seem the authors made a conscious decision to do this. No references are given for sequential algorithms or WHO recommendations; the authors should correct this. In addition, the authors should mention whether other studies on ACF used one of these sequential algorithms and how that might affect comparability of these previous results to their own.

Reviewer #2: Comments have been added on several places wherever required. Comments usually revolves around identifying the nature and the type of mining and dealing with the confounders like silicosis and asbestosis.

6. PLOS authors have the option to publish the peer review history of their article (what does this mean?). If published, this will include your full peer review and any attached files.

Reviewer #1: No

Reviewer #2: **Yes: **Shamim Mohammad, Ph.D.

---

## [Author Response · Author response to Decision Letter 0]

13 Feb 2021

We thank the reviewer for this point. We have done so to the best of our understanding.

2. Please state whether the data utilized in this study were de-identified/anonymised before access? The data used in the study was de-identified before accessing for analyses

3. We note that Figure 1 in your submission contains map images which may be copyrighted. We require you to either (a) present written permission from the copyright holder to publish this figure specifically under the CC BY 4.0 license, or (b) remove the figure from your submission: The Figure 1 contained in the manuscript is not a copyrighted map. This map was developed purposely for this manuscript using ArcGIS Map Software, shape district files freely available from the Ghana Health Service and data collated from the study dataset. 

Response to Reviewer 1 Comments

The study by Ohene et al. describe active case-finding (ACF) for TB in several mining communities in Ghana and presents novel data that have potentially wider relevance to ACF in other countries, including what types of screening should be performed and what kind of yields can be expected. The authors should also be commended for undertaking a study based on routine data collected by the national TB program which remains an underutilized resource. However, the manuscript has a number of significant issues that need to be resolved. We thank the reviewer for the kind words and appreciate the time taken to review the paper.

We have responded to the various issues raised and note that the comments and suggestions have gone a long way to improve the paper.

The first primary issue with the manuscript is that it fails to describe completely how the screening was performed in each of the 3 regions, 21 districts, and 78 communities included in the study. Without these additional data, it is difficult to know what the true denominator for the study was and how representatively the In particular, I was looking for additional details in the Results (lines 192-217) such as:

- What was the estimated catchment size of the outreach performed in each community? 

- What kind of catchment was expected as a result of the community outreach and radio announcements? (For example, in the South Africa Thibela TB study, Grant et al. described these activities in a separate paper, AIDS 2010 24:S37.)

- What were the demographics of these communities compared to the demographics of the study population (i.e., the people who showed up for screening)? We thank the reviewer for this important observation. 

We obtained the data for this study from the NTP after the program had undertaken the TB screening exercises in the selected artisanal mining communities in which mining activities are generally operated illegally. 

The objective of the TB screening exercise was to identify people with TB in these hard to reach artisanal mining areas deemed to be at high risk for TB but with limited access to TB services. 

The NTP exercise was not meant to be a study in which sampling or randomization into intervention and control communities was done. It was an exercise offering TB screening, testing and treatment to all community members 15 years and above who were willing to participate (after the mobilization/sensitization activities done prior to the screening) so those who voluntarily made themselves available were screened.

The NTP database had the total number of people who were screened in the communities but did not have data of number of people in the communities in which screening was conducted. 

We were therefore not able to estimate the catchment size of each community in which the outreach performed nor assess the demographics of the communities compared to the study population. 

We agree that depending on which people came forward, the population screened may not have been representative of the communities in which the exercise was undertaken creating the opportunity for a possible bias.

We have elaborated more on the context in which the screening exercise was undertaken in the Methods and highlighted the voluntary nature of the TB screening exercise and the fact that data on the community population was not available making it impossible to deduce proportion of the population screened.

We have also included the data limitations in the Limitations section of the Discussion so that readers will interpret the findings in the light of the possible biases.

- In addition, Table 1 presenting the demographics of the persons screened is difficult to interpret without the demographics of the regions to compare them against. These data ought to be available from census or modeled projects such as WorldPop. We thank the reviewer for the suggestion to include demographic data from the census information.

We have presented the age and sex of the regions using census data in Table 1 to enable comparison with the demographics of the population screened as suggested.

In the Results, we have compared the age and sex of the population screened with these demographics reported in census data. 

- Are there reasons to believe the sample was biased in any way? 

For example, did fewer men or mining workers show up than expected? 

If there was bias, was the bias different for each region, district, or community? We compared the study population data with census data as proxy for the proportions of male/female and age group in the community and noted that slightly fewer men showed up for the screening.

We also noted that those screened consisted of a relatively higher proportion of older people compared to what would be expected in the community using census data as proxy.

This general trend was noted across the 3 regions.

We recognize that, using census data as proxy and assuming that the community demographics are similar to the census data, the population screened may not necessarily reflect the community demographics and we have discussed this in the Discussion and highlighted this among the study limitations.

- The authors state that the people screened per community “ranged from 39 to 202” (line 194). Are these meaningful differences in terms of the community population sizes? We do not have information on the community population sizes limiting the ability to compare differences in terms of the community population sizes.

We have stated that the data set did not have this information.

As a result of the above exclusions, I find some of the claims and intended comparisons unsupported. For example, in several instances, the authors discuss the prevalence in the communities. For example, in the Results the authors state “the community prevalence varied between 0 and 6.8%” (line 232). We thank the reviewer for this point.

We have corrected the narrative to read “prevalence among the people screened” to accurately reflect the information being reported. 

Later, in the Discussion the authors compare these results on "the overall community TB prevalence" to published results in other countries such as South Africa and Zambia (lines 335-340). However, without a proper accounting of the sampling and any correction between the study population and the overall target population, these claims are unjustified. (For example, one can imagine a scenario where only the most ill mine workers showed up in one region with good outreach, while less ill mine workers and perhaps their families and friends showed up in another region.)

The authors should exercise care and qualify the language around these claims, in addition to explaining how screening was performed and what the demographics of the communities were. The authors make a partial acknowledgment late in the Discussion ("Comparing across such studies may however not be straightforward..." lines 368). However, by then the claims have already been made. We thank the reviewer for this important observation. In order to set the record straight from the start, we have indicated early on in the Discussion the context of our study and the point that it was not possible to tell whether the people screened fairly represented the rest of the community.

We have added that interpretations of the comparisons with other studies should be done against the knowledge of the background of our study setting.

In Table 3, describing the bivariate analyses, it is not clear what statistical test was performed to arrive at a single p-value per demographic variable per region. What test is this?

What is the null hypothesis in each case?

Why were the 3 regions analyzed separately rather than jointly in a hierarchical/mixed effects model?

Also, claims in this section do not appear supported by the results. For example, "Those with TB were more likely to be male" in line 261. What test and test statistic supports this?

- Table 4 presents the results of a logistic regression, but again here, the model used is unclear. The authors should state the logistic regression equation model in the Methods.

In addition, the accompanying text should clarify the difference in purpose between Table 3 and Table 4, as they appear to have the same function. We thank the reviewer for drawing our attention to missing information on the statistical test which was chi square test.

We have changed the focus of Table 3 which was previously bivariate analysis to simply show the regional distribution of TB cases across the demographic variables as some readers may be interested in the regional details. 

We have revised the wording in the Results to reflect this change.

For Table 4, we first conducted univariate logistic regression assess factors associated with TB diagnosis. 

Given the few characteristics being examined, all variables were then entered into a binomial logistic regression model with the outcome variable being a diagnosis of TB

- Table 5 is unclear in several aspects. What does "X-ray suggestive of TB" mean? Is this the same as, say, an X-ray with Xpert-confirmed TB? What is the purpose of this section (from line 278)? If it is to establish that particular symptoms may be as good as Xpert-confirmed X-ray screening, the authors should state so.

Also, how should the reader interpret the kappa statistic, compared to say the more familiar correlation coefficient? In Table 5, we are assessing the strength of association solely between symptoms and X-ray findings suggestive of TB alone. This analysis has no relation to diagnosis of Xpert confirmed TB. 

We have simplified that section to make the purpose clearer and deleted the logistic regression analysis results from Table 5 and the narrative.

Instead of using Cohen’s kappa statistic as a measure of agreement, we have revised the statistic to Phi correlation coefficient since we are assessing strength of association between 2 dichotomous variables the presence of particular symptoms suggestive of TB and X-ray suggestive of TB.

The interpretation of the r value as follows .01 - .19 No negligible relationship, .20 -.29 Weak positive relationship, .30 -.39 Moderate positive relationship, .40 - .69 Strong positive relationship, .70 or higher very strong positive relationship. 

Finally, the authors bring up a good point at the end of the Discussion that should be raised earlier, comparing combined symptom and X-ray screening compared to a sequential algorithm. In particular, the former seems faster and perhaps easier, and it would seem the authors made a conscious decision to do this. 

No references are given for sequential algorithms or WHO recommendations; the authors should correct this. In addition, the authors should mention whether other studies on ACF used one of these sequential algorithms and how that might affect comparability of these previous results to their own. We thank the reviewer for the comment which we have addressed. 

We have also provided the reference for the sequential algorithms.

We have also mentioned other studies and addressed comparability of results as suggested. 

Response to Reviewer 2 Comments

Points Raised Response

Reviewer #2: Comments have been added on several places wherever required. Comments usually revolves around identifying the nature and the type of mining and dealing with the confounders like silicosis and asbestosis. We thank the reviewer for the comments.

We also note with gratitude that addressing the comments has helped to improve the paper.

Line 121

Introduction doesn't discuss about what kind of mining communities. This is pertinent question to address. Whether these communities are involved in sandstone, granite, marble, asbestos etc. mining. If this is the case, the diagnosis gets very technical. Because, the cases of silicosis and asbestosis have the similar symptoms as the TB cases.

Therefore,

There is a need to explain the word mining communities

Sandstone miners mostly suffers from the silicosis and are commonly misdiagnosed as the TB cases We have taken note of the comment with appreciation. We have now provided more information on the kind of mining communities in the Methods highlighting that they are small scale artisanal mining communities with the miners undertaking illegal gold mining activities. 

In our study TB diagnosis was solely based on a positive GenXpert result. 

We acknowledge that silicosis may commonly be misdiagnosed as TB, however we only included bacteriologically confirmed TB using GenXpert.

Line 140Is it possible to discriminate between the cases of TB, Silicosis or asbestosis only by X Rays? It was not possible to distinguish between the cases of TB, silicosis or asbestosis only by X Rays.

The TB cases that we report in our study were bacteriologically confirmed using GenXpert.

The X-ray was used as a screening method. Those reported to have an abnormal X-ray suggestive of TB were among those requested to produce sputum to be tested for TB using GenXpert.

We have also included in the Discussion that it is important to bear in mind that those who had symptoms suggestive of TB or were reported to have X-ray suggestive of TB could also have had silicosis since symptoms and X-ray findings may be similar in both conditions.

Line 148

Nothing randomized in the method section, be it area selection or the selection of the community. All selected purposively. In such scenario-----how to address generalization component of this research?? We agree that the communities targeted for screening were purposefully selected based on the presence of informal mining activities and the population screened came voluntarily for the exercise and so in the absence of randomization and an indication of population coverage, it may be difficult to generalize the study findings to other areas.

We have acknowledged this limitation of the study and point out that due to the selective nature of the study population, the findings may not be generalizable to other settings.

Line 156

Similar symptoms are also there with the miners with the silicosis We acknowledge that similar symptoms are found in miners with silicosis. The symptom screening was used to identify eligible people to produce sputum for TB testing. 

Even though some of the people having these symptoms may have had silicosis, the outcome of interest for the screening was a TB diagnosis which was done using GenXpert.

Line 406

Silicosis may be a confounder in the mining communities. Do you agree?? In our study, the outcome variable was a diagnosis of TB in our study which was bacteriologically confirmed using GenXpert.

---

## [Decision Letter · Decision Letter 1]

4 Mar 2021

Case finding of tuberculosis among mining communities in Ghana

PONE-D-20-34085R1

Dear Dr. Ohene,

We’re pleased to inform you that your manuscript has been judged scientifically suitable for publication and will be formally accepted for publication once it meets all outstanding technical requirements.

Kind regards,

Frederick Quinn

Academic Editor

PLOS ONE

Additional Editor Comments (optional):

Reviewers' comments:

Reviewer's Responses to Questions

**Comments to the Author**

1. If the authors have adequately addressed your comments raised in a previous round of review and you feel that this manuscript is now acceptable for publication, you may indicate that here to bypass the “Comments to the Author” section, enter your conflict of interest statement in the “Confidential to Editor” section, and submit your "Accept" recommendation.

Reviewer #2: All comments have been addressed

2. Is the manuscript technically sound, and do the data support the conclusions?

Reviewer #2: Yes

3. Has the statistical analysis been performed appropriately and rigorously? 

Reviewer #2: Yes

4. Have the authors made all data underlying the findings in their manuscript fully available?

Reviewer #2: Yes

5. Is the manuscript presented in an intelligible fashion and written in standard English?

Reviewer #2: Yes

6. Review Comments to the Author

Reviewer #2: Authors have addressed all the queries comprehensively They have defined the type and nature of the mining and addressed the question of confounders.

7. PLOS authors have the option to publish the peer review history of their article (what does this mean?). If published, this will include your full peer review and any attached files.

Reviewer #2: **Yes: **Shamim Mohammad, M.Phil., Ph.D.

---

## [Editor Report · Acceptance letter]

8 Mar 2021

PONE-D-20-34085R1 

Case finding of tuberculosis among mining communities in Ghana 

Dear Dr. Ohene:

I'm pleased to inform you that your manuscript has been deemed suitable for publication in PLOS ONE. Congratulations! Your manuscript is now with our production department. 

Kind regards, 

on behalf of

Dr. Frederick Quinn 

Academic Editor

PLOS ONE